# Dome-shaped magnetic order competing with high-temperature superconductivity at high pressures in FeSe

J.P. Sun[1,*], K. Matsuura[2,*], G.Z. Ye[1,3,*], Y. Mizukami[2], M. Shimozawa[4], K. Matsubayashi[5], M. Yamashita[4], T. Watashige[6], S. Kasahara[6], Y. Matsuda[6], J.-Q. Yan[7,8], B.C. Sales[7], Y. Uwatoko[4], J.-G. Cheng[1] & T. Shibauchi[2]

The coexistence and competition between superconductivity and electronic orders, such as spin or charge density waves, have been a central issue in high transition-temperature ($T_c$) superconductors. Unlike other iron-based superconductors, FeSe exhibits nematic ordering without magnetism whose relationship with its superconductivity remains unclear. Moreover, a pressure-induced fourfold increase of $T_c$ has been reported, which poses a profound mystery. Here we report high-pressure magnetotransport measurements in FeSe up to $\sim 15\,GPa$, which uncover the dome shape of magnetic phase superseding the nematic order. Above $\sim 6\,GPa$ the sudden enhancement of superconductivity ($T_c \leq 38.3\,K$) accompanies a suppression of magnetic order, demonstrating their competing nature with very similar energy scales. Above the magnetic dome, we find anomalous transport properties suggesting a possible pseudogap formation, whereas linear-in-temperature resistivity is observed in the normal states of the high-$T_c$ phase above 6 GPa. The obtained phase diagram highlights unique features of FeSe among iron-based superconductors, but bears some resemblance to that of high-$T_c$ cuprates.

[1] Beijing National Laboratory for Condensed Matter Physics and Institute of Physics, Chinese Academy of Sciences, Beijing 100190, China. [2] Department of Advanced Materials Science, University of Tokyo, Kashiwa, Chiba 277-8561, Japan. [3] School of Physical Science and Technology, Yunnan University, Kunming 650091, China. [4] The Institute for Solid State Physics, The University of Tokyo, Kashiwa, Chiba 277-8581, Japan. [5] Department of Engineering Science, The University of Electro-Communications, Chofu, Tokyo 182-8585, Japan. [6] Department of Physics, Kyoto University, Sakyo-ku, Kyoto 606-8502, Japan. [7] Materials Science and Technology Division, Oak Ridge National Laboratory, Oak Ridge, Tennessee 37831, USA. [8] Department of Materials Science and Engineering, University of Tennessee, Knoxville, Tennessee 37996, USA. * These authors contributed equally to this work. Correspondence and requests for materials should be addressed to J.-Q.Y. (email: jqyan@utk.edu) or to J.-G.C. (email: jgcheng@iphy.ac.cn) or to T.S. (email: shibauchi@k.u-tokyo.ac.jp).

                                                 

In most iron-based superconductors, superconductivity emerges on the verge of a long-range antiferromagnetically ordered state[1], which is a common feature to many unconventional superconductors[2,3] including the cuprates and heavy-fermion materials. It has been shown that the antiferromagnetic order in the iron-pnictide materials accompanies or follows the tetragonal-to-orthorhombic structural transition at $T_s$. In striking contrast, the structurally simplest FeSe exhibits a high $T_s \approx 90$ K but no magnetic order appears at lower temperatures[4–7], and still its ground state is an unconventional superconducting state with $T_c \approx 9$ K (refs 8–10). This material is also intriguing in that in the form of one-unit-cell-thick films a very high $T_c$ (up to 109 K) has been reported recently[11–13], which is likely associated with a carrier-doping effect[14,15] from the substrate. In bulk FeSe, a significant electronic anisotropy is found below $T_s$ in the nonmagnetic orthorhombic phase, which is often called a nematic state[9,10,16,17]. In the nematic phase, very small Fermi surfaces with strong deviations from the first-principles calculations have been observed[10,18,19], and the occurrence of superconductivity with such small Fermi energies is quite unusual, implying that the system is deep in the crossover regime between the weak-coupling Bardeen–Cooper–Schrieffer and strong-coupling Bose–Einstein–condensate limits[10].

In addition to these distinct electronic characteristics of FeSe, remarkable properties have been reported under high pressure[20–27]. First of all, the initial study on powder samples has shown that the relatively low $T_c \approx 9$ K at ambient pressure can be enhanced by more than fourfold to $\sim 37$ K under $\sim 8$ GPa, pushing it into the class of high-$T_c$ superconductors[21]. More recent studies under better hydrostatic pressure conditions revealed a complex temperature–pressure ($T$–$P$) phase diagram featured by a suppression of $T_s$ around 2 GPa, a sudden development of static magnetic order above $\sim 1$ GPa (ref. 23), and an enhancement of $T_c$ in a three-plateau process[24], that is, $T_c \sim 10(2)$ K for 0–2 GPa, $T_c \sim 20(5)$ K for 3–5 GPa, and $T_c \sim 35(5)$ K for 6–8 GPa. The first jump of $T_c$ from $\sim 10$ to $\sim 20$ K seems to coincide with the suppression of the nonmagnetic nematic state and the development of the long-range magnetic order at $T_m$ evidenced by μSR measurements[22]. The observation that both $T_c$ and $T_m$ increase with pressure in the pressure range 1–2.5 GPa has been taken as evidence for the cooperative promotion of superconductivity by the static magnetic order[22]. Such a scenario, however, does not fit into the general scope of iron-based superconductors, in which the optimal superconductivity is realized when the long-range magnetic order is close to collapse[1,28]. This issue remains unclear unless the fate of magnetic order at $T_m$ under higher pressures is sorted out. Due to the technical limitations of probing small-moment magnetic order above 3 GPa, this task only becomes possible very recently when a clear signature at $T_m$ is visible in the resistivity[23,25,26] of high-quality FeSe single crystals[29]. We also note that more recently the pressure-induced magnetic order in these single crystals has been confirmed below $T_m$ by Mössbauer[30] and nuclear magnetic resonance (NMR) measurements[31].

Here by performing the high-pressure resistivity $\rho(T)$ measurements up to $\sim 15$ GPa on high-quality single crystals, we construct for bulk FeSe the most comprehensive $T$–$P$ phase diagram mapping out the explicit evolutions with pressure of $T_s$, $T_c$ and $T_m$. We uncover a previously unknown dome-shaped $T_m(P)$, having two end points situated on the boundaries separating the three plateaus of $T_c(P)$. Our results thus provide compelling evidence linking intimately the sudden enhancement of $T_c$ to 38 K to the suppression of long-rang magnetic order. This highlights a competing nature between magnetic order and high-$T_c$ superconductivity in the phase diagram of FeSe, which is a key material among the iron-based superconductors.

## Results

**Low-pressure region.** The tetragonal-orthorhombic structure transition at $T_s \approx 90$ K for bulk FeSe (blue square in Fig. 1) is manifested as a slight upturn in resistivity, which can be taken as a signature to track down the evolution of $T_s$ with pressure. Our resistivity $\rho(T)$ data measured with a self-clamped piston–cylinder cell (PCC) up to $\sim 1.9$ GPa are shown in Fig. 2a. As can be seen, $T_s$ is suppressed progressively to below 50 K at $\sim 1.5$ GPa, above which the anomaly at $T_s$ becomes poorly defined. Meanwhile, a second anomaly manifested as a more profound upturn in $\rho(T)$ emerges at $T_m \sim 20$ K and moves up steadily with pressure. In light of the recent high-pressure μSR, Mössbauer, and NMR studies[22,30,31], this anomaly at $T_m$ corresponds to the development of long-range magnetic order. We also note that in this magetically ordered state below $T_m$, the orthorhombic structure similar to the one (space group $Cmma$) in the nematic phase has been reported recently[30,31]. $T_s$ and $T_m$ seem to cross around $\sim 2$ GPa. In this pressure range, the superconducting transition temperature $T_c$ (defined as the zero-resistivity temperature) first increases and then decreases slightly before rising again. This features a small dome-shaped $T_c(P)$ peaked at $\sim 1.2$ GPa (Fig. 1), which roughly coincides with the pressure where the long-range magnetic order at $T_m$ starts to emerge. These results in this relatively low-pressure range are in general consistent with those reported previously[23,25,26].

**High-pressure region.** To further track down the evolution of $T_m$, we turn to $\rho(T)$ measurements in cubic anvil cells (CACs) that can maintain a quite good hydrostaticity up to $\sim 15$ GPa (refs 32–34). Figure 2b–d displays the $\rho(T)$ data measured in two self-clamped CACs and one constant-loading CAC (see Methods for experimental details). In line with the results of PCC in Fig. 2a, the sudden upturn is clearly visible at $T_m$ in both measurements in the pressure range up to $\sim 2.5$ GPa (Figs 2b,d and 3a,b), above which the upturn anomaly disappears and instead a kink appears in $\rho(T)$ followed by a gradual drop before reaching the superconducting transition (Figs 2b,d and 3c,d). Our

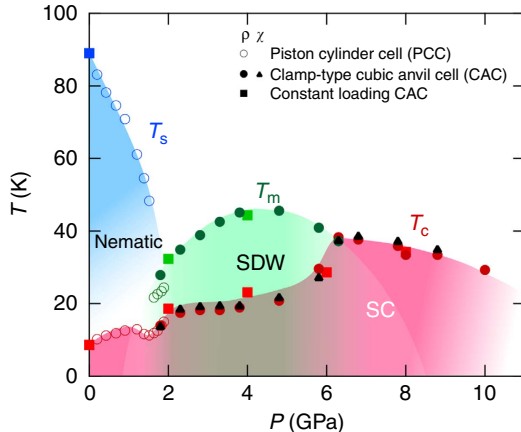

**Figure 1 | Temperature–pressure phase diagram of bulk FeSe.** The structural ($T_s$, blue), magnetic ($T_m$, green), and superconducting transition temperatures ($T_c$, red and black) as a function of hydrostatic pressure in high-quality single crystals determined by anomalies in resistivity $\rho(T)$ measured in the PCC (open circles), clamp-type CAC (closed circles), and constant-loading type CAC (closed squares). $T_c$ values determined from the ac magnetic susceptibility ($\chi(T)$) measurements in the clamp-type CAC are also shown (solid triangles). The magnetic phase is most likely a spin density wave (SDW) phase. Colour shades for the nematic, SDW, and superconducting (SC) states are guides to the eyes.

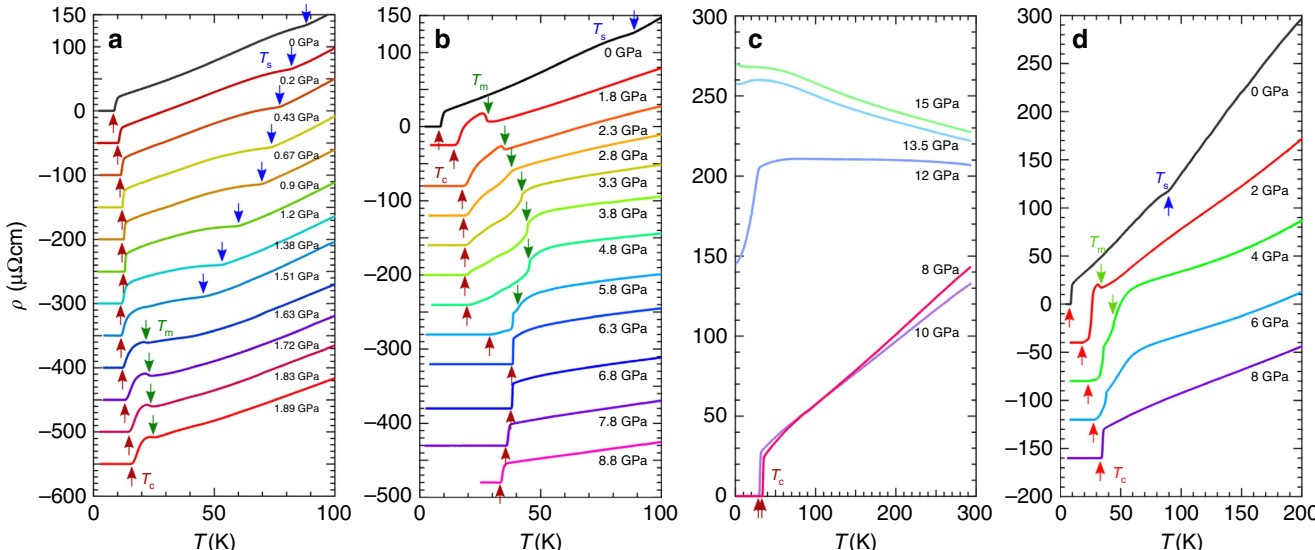

**Figure 2 | Temperature dependence of resistivity in FeSe single crystals under high pressure.** (**a**) $\rho(T)$ curves below 100 K at different pressures up to ∼1.9 GPa measured in the self-clamped PCC. (**b**) Data up to ∼8.8 GPa measured in the self-clamped CAC. (**c**) Data up to ∼15 GPa measured in the smaller self-clamped CAC. (**d**) $\rho(T)$ curves below 200 K up to 8 GPa measured in the constant-loading CAC. Except for (**c**) the data are vertically shifted for clarity. The resistive anomalies at transition temperatures $T_s$, $T_m$, and $T_c$ are indicated by the arrows.

self-clamped CAC enables us to measure $\rho(T)$ under different magnetic fields[32,33], which clearly resolves that this anomaly is of the same origin as the magnetic order at lower pressures. As seen in Fig. 4b, on the application of magnetic fields the resistivity kink at 2.8 GPa restores to the upturn anomaly and this anomaly shows little field dependence as that observed at 1.8 GPa (Fig. 4a). On further increasing pressures, $T_m$ moves up gradually and reaches about 45 K at 4.8 GPa, where the resistivity kink becomes sharper (Fig. 2b). In striking contrast with the progressive enhancement of $T_m$ in this pressure range of 2–5 GPa, $T_c$ remains nearly unchanged at the second plateau of ∼20 K.

The situation changes markedly when further increasing pressure above ∼5 GPa. As shown in Figs 2b and 3d, $\rho(T)$ at 5.8 GPa exhibits an abrupt drop at 38.5 K before reaching zero resistivity at $T_c = 27.5$ K, and then at 6.3 GPa the abrupt drop develops into a very sharp superconducting transition at $T_c = 38.3$ K with a transition width of only 0.4 K (Figs 2b and 3e). Thus, $T_c$ is nearly doubled from ∼20 K at 4.8 GPa to 38.3 K at 6.3 GPa, and the abrupt drop at 38.5 K under 5.8 GPa corresponds to the onset of superconductivity rather than the magnetic order, which is supported by the large change under magnetic fields (Fig. 4d). Then, what is the fate of $T_m$? A closer inspection of the $\rho(T)$ data at 5.8 GPa reveals an inflection point at the temperature slightly above the abrupt drop. This feature can be well resolved from the temperature derivative of resistivity $d\rho/dT$ in Fig. 3d. Here the small peak centred at 41 K corresponds to $T_m$, which is confirmed by the field independence (Fig. 4d). At 6.3 GPa, such a magnetic anomaly is absent in the $d\rho/dT(T)$ curve at zero field, but becomes visible under 9-T field when $T_c$ is suppressed to lower temperatures, as shown in Figs 3e and 4i. On further increasing pressure, the superconducting transition remains very sharp and $T_c$ moves down slowly to 33.2 K at 8.8 GPa. Due to the large upper critical field, $T_m$ cannot be defined at $P > 6.3$ GPa (Fig. 4j–l). These above findings are further confirmed by separate high-pressure resistivity measurements with a constant-loading CAC, as shown in Fig. 2d.

At higher pressures, the resistivity curves measured by using a small CAC (Fig. 2c) show a sudden change from metallic and superconducting behaviour to semiconducting and

non-superconducting one at around 12 GPa. Indeed, the crystal structure studies of FeSe polycrystals at high pressures[35,36] have shown that high pressures above ∼10 GPa will stabilize the collapsed three-dimensional orthorhombic *Pbnm* structure, which is non-superconducting. Therefore, we will focus our attention to the phase diagram at $P < 12$ GPa, where the crystal structure symmetries should be the same as the ones in the ambient pressure.

The superconductivity is also checked by the ac susceptibility measurements under pressure using the self-clamped CAC (Fig. 5). The onset temperature $T_c^\chi$ of the diamagnetic signal is quantitatively consistent with the zero-resistivity $T_c$ at each pressure (Fig. 1). Moreover, the magnitude of the diamagnetic signals at lowest temperatures does not show significant pressure dependence, suggesting the bulk superconducting nature in a wide pressure range covering the nematic, magnetic and nonmagnetic phases.

## Discussion

When the obtained $T_m(P)$ and $T_c(P)$ data are plotted in Fig. 1, we uncover a previously unknown dome-shaped magnetic order with two ends situated near the boundaries separating the three-plateau $T_c(P)$. This observation suggests that the superconductivity in FeSe is intimately correlated with the magnetism. Despite the absence of static magnetic order within the nematic phase at ambient pressure[5–7], strong spin fluctuations with an in-plane wave vector $\mathbf{q} = (\pi, 0)$ have been identified recently[37]. Recent theoretical studies[38] have attributed the absence of static magnetic order to unusual magnetic frustration among multiple competing magnetic orders with $\mathbf{q} = (\pi, \xi)$ with $0 \le \xi \le \pi/2$. According to the calculations, this magnetic frustration can be removed under pressure, and gives way to a long-range magnetic order with $\mathbf{q} = (\pi, 0)$. Indeed, the stabilization of static magnetic order at $T_m$ is concomitant with the destruction of nematic order as seen in Fig. 1, thus highlighting the competing nature between nematic and magnetic order. The $\mathbf{q} \ne 0$ nature of the pressure-induced magnetic phase is supported by the Fermi surface reconstruction recently reported from quantum oscillations[25], and also

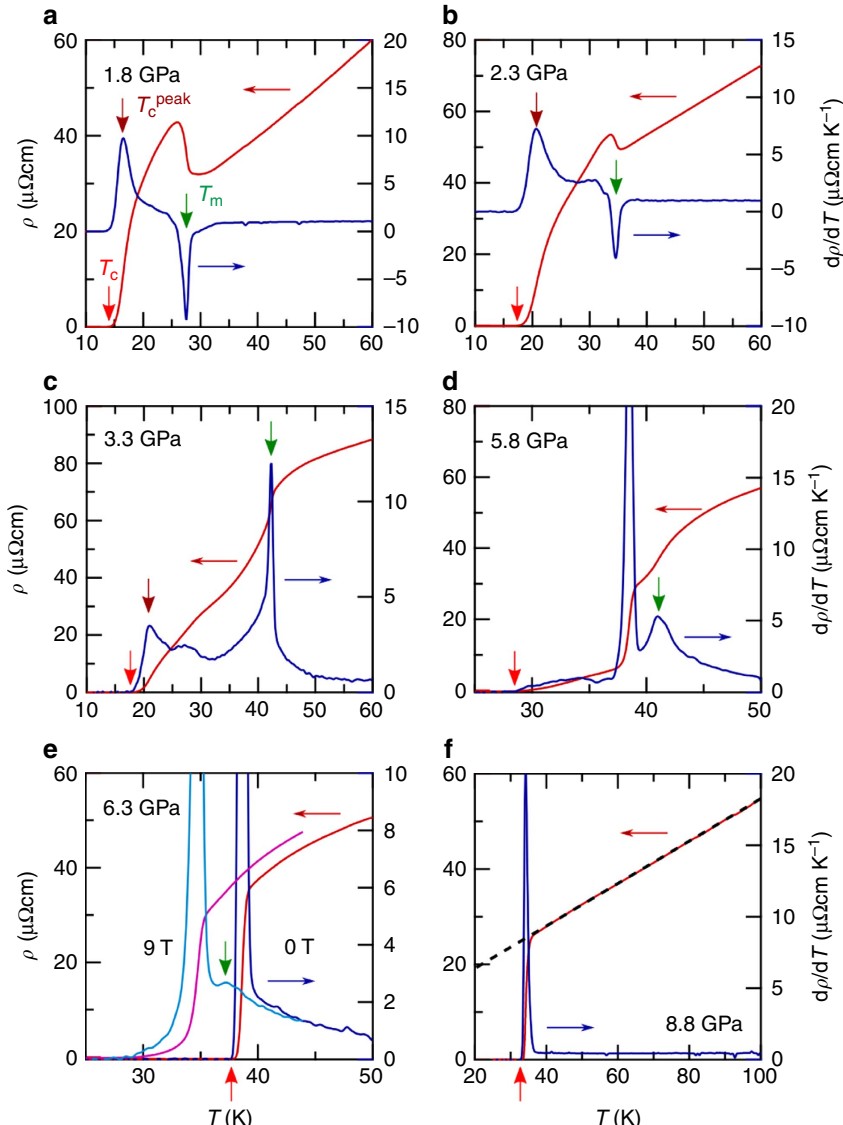

**Figure 3 | Determination of magnetic and superconducting transition temperatures from the resistivity curves under high pressure. (a–f)** Temperature dependence of resistivity $\rho(T)$ (left axis) and temperature derivative of resistivity $d\rho/dT(T)$ (right axis) at low temperatures. The temperature at the maximum slope of superconducting transition is defined as $T_c^{peak}$ and the zero resistivity temperature as $T_c$. The magnetic transition temperature $T_m$ is determined by the dip or peak in $d\rho/dT(T)$. At 6.3 GPa, the data at 9 T is also shown where $T_m$ anomaly appears when the superconducting transition is suppressed to a lower temperature (**e**). At 8.8 GPa, the normal-state resistivity can be fitted to a $T$-linear dependence (dashed line in (**f**)).

consistent with the field independence of $T_m$ found in the present study (Fig. 4a–d). Thus the pressure-induced order that supersedes the nematic order above $\sim 2$ GPa is most likely a spin density wave (SDW), although the previous neutron scattering experiment was unsuccessful due to the small magnetic moment[22]. The most recent NMR study as well as the orthorhombicity found below $T_m$ also support the stripe-type SDW order with $\mathbf{q} = (\pi, 0)$[30,31]. Most importantly, our results reveal unambiguously that the sudden jump of $T_c$ around 6 GPa comes concomitantly with the suppression of magnetic order, demonstrating another competition between the SDW order and high-$T_c$ superconductivity at high pressures. The peak temperature ($\approx 45$ K) of the $T_m(P)$ dome is quite close to the maximum $T_c (= 38.3$ K), indicating that these competing orders have very similar energy scales. This is different from other iron-based superconductors where the antiferromagnetic transition temperatures in the mother materials are significantly higher than the maximum $T_c$ values of the derived superconducting

phases[1,28]. We note that the non-monotonic dependence of $T_c(P)$ is somewhat akin to the two-dome shape of $T_c(x)$ recently found in LaFeAsO$_{1-x}$H$_x$ (refs 1,39). However, the complete dome shape of $T_m(P)$ with two ends in a single-phase diagram of FeSe without changing of carrier balance is distinctly different from the case of LaFeAsO$_{1-x}$H$_x$, in which two different antiferromagnetic orders exist near the opposite doping ends of the superconducting phase. Altogether with the presence of competing nematic order at low pressures, this phase diagram of FeSe thus exhibits unique features among iron-based superconductors. It should be noted that a phase diagram bearing some resemblance has been recently found in hole-doped cuprate superconductors, where the high-$T_c$ superconducting dome is partially suppressed by the presence of dome-shaped charge density wave (CDW) order[3].

It is also intriguing to point out that above the SDW dome, the resistivity curves in the pressure range of $\sim 3$–6 GPa show anomalous sub-linear (convex) temperature dependence

**Figure 4 | Effects of magnetic fields on the magnetic and superconducting transitions under high pressure.** (**a–d**, **i–l**) $\rho(T)$ curves at different magnetic fields applied parallel to the $c$ axis. The magnetic transition temperature $T_m$ is field independent and marked by the green arrow (**a–d**). At 6.3 GPa, the $T_m$ anomaly is only visible at high fields (**i**). (**e–h**, **m–p**) $H$-$T$ phase diagrams. The zero resistivity is attained below $T_c$ and the $T_c^{peak}$ line (determined by the peak in $d\rho/dT(T)$) is a lower bound of upper critical field $H_{c2}$. The sharp superconducting transitions under magnetic fields for $P > 6.3$ GPa indicate narrowed regimes of the vortex liquid state.

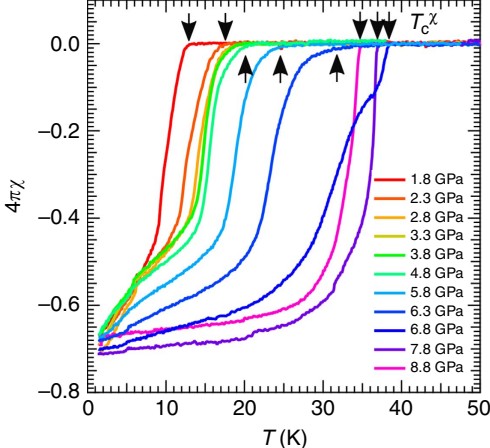

**Figure 5 | Temperature dependence of ac susceptibility $\chi$ under high pressure.** The data are taken by using the clamp-type CAC. The onset temperatures $T_c^\chi$ of diamagnetic signals are marked by arrows.

(Fig. 2b,d), which mimics the suppression of the quasiparticle scattering by the pseudogap formation above the CDW phase of underdoped cuprates. At higher pressures, this convex anomaly in the temperature dependence of resistivity becomes less pronounced. At 8.8 GPa, $\rho(T)$ displays a perfect linear-in-$T$ dependence in a wide temperature range of the normal state (Figs 2b and 3f), which is a hallmark of non-Fermi-liquid behaviour. The observation of the non-Fermi-liquid behaviour at high pressures where the magnetic order is vanishing is an indication of the presence of strong critical fluctuations. Similar $T$-linear $\rho(T)$ above $T_c$ has been observed near a quantum critical point (QCP) at $x \approx 0.3$ in the $BaFe_2(As_{1-x}P_x)_2$ system[28]. In that system, the critical fields near the QCP show anomalous features pointing to an enhancement of the energy of superconducting vortices possibly due to a microscopic mixing of antiferromagnetism and superconductivity[40]. The analysis of the upper critical field $H_{c2}$, which is related to the effective mass $m^\star$ in a simple picture by $m^{*2} \propto -(1/T_c)dH/dT|_{T=T_c}$ (ref. 41), has been shown to be much less sensitive to the mass enhancement observed in other techniques near the QCP. In our similar

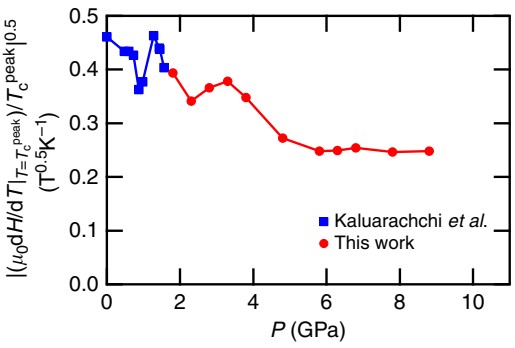

**Figure 6 | Mass analysis from H–T curves under high pressure.** The pressure dependence of the square root of the initial slope of the $T_c^{peak}$ line divided by $T_c^{peak}$, which corresponds to the effective mass in a simple picture. For comparison, the low-pressure data from ref. 26 are also indicated (blue squares).

analysis (Fig. 6) we also find a rather flat behaviour at high pressure. This suggests that the vortex state of FeSe at high pressures is also nontrivial.

We also note that the unusually strong vortex pinning has been found in clean films of $BaFe_2(As_{1-x}P_x)_2$ near the QCP[42]. Our H-T phase diagrams at different pressures indicate that the vortex-liquid region between the width of transition ($T_c$ and $T_c^{peak}$) shows a dramatic narrowing when the SDW is suppressed below the superconducting transition (Fig. 4e–h, m–n), which can also be accounted for by the enhanced pinning near the QCP. However, a peculiar feature here is that the optimal $T_c$ takes place at the point where $T_m$ just crosses $T_c$, rather than the possible QCP at $\gtrsim 8$ GPa where $T_m$ completely vanishes. This is again rather similar to the case of hole-doped cuprate superconductors in a sense that a putative QCP is located at the slightly overdoped side[3].

To summarize, by using high pressure as a clean tuning knob, we have clarified in bulk FeSe the interplay of three competing orders, that is, nematic, magnetic and superconductivity, and elucidated how the high-$T_c$ superconducting phase is achieved. The relatively low $T_c$ at ambient pressure should be attributed to the presence of competing nematic order; the application of high pressure initially suppresses gradually the nematic order so as to enhance superconductivity. At the same time, the suppression of nematic order also relieves the magnetic frustration so as to stabilize the magnetic order, which then competes again with superconductivity, leading to a local maximum of $T_c(P)$ around 1 GPa. Then, a complete elimination of nematic order around 2 GPa gives rise to a more profound enhancement of $T_c$ to the second plateau of $\sim 20$ K; the opposite effects on $T_c$ imposed by the strong fluctuations of nematic order and the long-range SDW order produce the local minimum of $T_c(P)$ around 1.5 GPa. Above $\sim 2$ GPa, only the static SDW order is present and competes with superconductivity so that $T_c$ keeps nearly unchanged while $T_m$ rises until $\sim 5$ GPa, where the magnetic order at $T_m$ eventually becomes destabilized by pressure. Then, the suppression of magnetic order leads to a great enhancement of $T_c$ to 38 K. In such a way, FeSe exhibits a unique phase diagram with three competing orders, nematic, SDW and high-$T_c$ superconductivity; the latter two of which have very close energy scales. This newly constructed diagram, which shares some similarities to that of cuprates, may offer important clues for discussing the unconventional origins of the high-$T_c$ superconductivity in this class of materials.

## Methods

**Sample preparation and characterization.** High-quality FeSe single crystals used in the present study have been grown by two different methods: the chemical vapour transport technique[29] (Kyoto University) and the flux method[43] (Oak Ridge National Laboratory). These two methods yield FeSe single crystals with similar quality in terms of the phase transition temperatures ($T_s = 87$–90 K and $T_c = 8.5$–9 K) and the residual resistivity ratio RRR $\sim 40$. However, the flux method produces much larger (up to $\sim 10$ mm) and thicker ($\sim 1$ mm) crystals and thus we cleaved and cut these samples to fit in the pressure cells. The crystals are well characterized by means of X-ray diffraction, energy dispersive spectroscopy, magnetic and transport properties under ambient pressure.

**High-pressure measurements.** High-pressure resistivity $\rho(T, P)$ measurements have been performed under hydrostatic pressures up to $\sim 2$ GPa with a PCC and up to $\sim 15$ GPa with CACs, respectively. For the crystals grown with the flux method, $\rho(T, P, H)$ data under high pressures and magnetic fields were measured in the Institute of Physics, Chinese Academy of Sciences, by using the self-clamped type CACs and PCC. The data up to 9 GPa have been taken by a 4-mm CAC, in which we also measured ac susceptibility by a conventional mutual inductance technique. Higher-pressure data up to 15 GPa were taken by a 2.5-mm CAC. The high-pressure resistivity of crystals grown with the chemical vapour transfer method was measured in the Institute for Solid State Physics, University of Tokyo with a constant-loading type CAC, which can maintain a nearly constant pressure over the whole temperature range from 300 to 2 K. The pressure value inside the PCC was determined by monitoring the shift of superconducting transition of lead (Pb), while those of CACs were calibrated at room temperature by observing the characteristic transitions of bismuth (Bi). (It should be noted that for the self-clamped pressure cells, both PCC and CAC, the pressure value at room temperature is slightly different from that at low temperature due to the solidification of liquid pressure transmitting medium and the different thermal contraction of cell components. Therefore, some corrections have been made in comparison with the data obtained from the constant-loading CAC). For all these high-pressure resistivity measurements, we employed glycerol as the pressure transmitting medium. All resistivity measurements were performed with the conventional four-terminal method with current applied within the ab plane and magnetic field perpendicular to the ab plane.

**Data availability.** The data that support the findings of this study are available on request from the corresponding authors (J.-Q.Y., J.-G.C., or T.S.).

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

## Acknowledgements

We thank T. Terashima, X.J. Zhou, T. Xiang, R. Yu, Q.M. Zhang, G. Chen, and J.-S. Zhou for very helpful discussions. We also thank Bosen Wang for his technical help. Work at IOP CAS was supported by the National Basic Research Program of China (Grant No. 2014CB921500), National Science Foundation of China (Grant No. 11574377, 11304371), and the Strategic Priority Research Program of the Chinese Academy of Sciences (Grant No. XDB07020100) as well as the Opening Project of Wuhan National High Magnetic Field Center (Grant No. 2015KF22), Huazhong University of Science and Technology. Work in Japan was supported by Grant-in-Aids for Scientific Research (A), (B), (S), and on Innovative Areas 'Topological Materials Science'. Work at ORNL was supported by the US Department of Energy, Office of Science, Basic Energy Sciences, Materials Sciences and Engineering Division.

## Author contributions

J.-Q.Y., J.-G.C. and T.S. conceived this project. J.P.S., G.Z.Y., J.-G.C. performed the high-pressure magneto-transport measurements with the self-clamped PCC and CAC; K.Matsuura, Y.Mizukami, M.S., K.Matsubayashi, M.Y., Y.U. measured the high-pressure resistivity with the constant-loading CAC; J.-Q.Y. and B.C.S. synthesized the FeSe single crystals with the flux method; K.Matsuura, T.W., S.K., Y.Matsuda grew the FeSe single crystals with the vapor transport technique. All authors discussed the results. J.-G.C. and T. S. wrote the paper with inputs from all authors. J.-G.C., Y.Matsuda, Y.U. and T. S. supervised the project.

## Additional information

**Competing financial interests:** The authors declare no competing financial interests.

