## [Peer Review File · Nature Communications]

Reviewers' Comments:

Reviewer #1 (Remarks to the Author)

The correlation between superconducting transition temperature T_c and electronic or spin orders such as spin or charge density wave or nematic ordering is the most important issue in both cuprate and Fe based superconductors. Therefore, the construction of phase diagrams for T_c in relation to a number of parameters such as charge carrier concentrations, magnetic transition temperatures, crystal lattice and symmetry, pressure or magnetic field is crucial to uncover what is closely linked to T_c . Among Fe-superconductors, FeSe superconductor's T_c in relation to magnetism has been unclear. By performing transport study at high pressure using three different pressure approaches, the authors have observed an evolution of T_c in relation to T_s (nematic transition) and magnetic transition, T_m . The authors constructed phase diagram which shows that dome-shaped magnetic order competes with the superconductivity of FeSe under high pressure. If the diagram reflects the T_c 's intrinsic correlation with T_s or T_m , it would be quite interesting, unique, and useful for further understanding of the high T_c superconductivity in FeSe. To achieve the intrinsic correlation, ideally, the crystal structures or crystal symmetry should remain unchanged at various high pressures. According to previous studies (EPL, 99 (2012) 26002; J Phys Chem B. 2010 Oct 7;114(39):12597-606. doi: 10.1021/jp1060446. Etc), high pressure can cause a few crystal structural or symmetry changes in FeSe.

The following are comments and suggestions:

- 1) The authors need to take into account the pressure induced possible structural changes and discuss how they can affect the phase diagram.
- 2) The authors define the T_c using zero resistance temperature. However, the superconducting transition temperature (T_c) seems more appropriate than $T_c(\text{zero})$ as a transition T_c may appear far above zero resistance T_c . For example, in Fig.2 b, the T_c (zero resistance is low), however, a transition T_c may already appear between T_m and T_c (zero resistance) for pressures from 2.8-4.8 GPa, which becomes very clear for 5.8 GPa. If this is the case, the shape of T_c vs pressure could be quite different in the phase diagram (Fig.1).
- 3) To clarify if the transition T_c is real, magnetic measurement under high pressure seems necessary.
- 4) The long tail of resistance from 2.8 to 5.8 GPa in Fig.2 may indicate the coexistence of secondary phase induced by pressure.
- 5) The selenium concentrations could be different for crystals made by CVD and flux growth method. This can give different crystal structural changes under high pressure.
- 6) If possible, crystal structural determination under the same pressure range should be done for the same crystal samples. This is critical to find out if it is the structural change or T_m induced by pressure that is responsible for the changes of T_c in FeSe.

Reviewer #2 (Remarks to the Author)

This is a good piece of the experimental work in which the authors have discovered the

dome-shaped structure of the magnetic phase in FeSe superconductor. They have shown that magnetism disappears for pressures exceeding 6GPa, and that such disappearance is associated with the sudden increase of the superconducting transition temperature T_c up to $\sim 38\text{K}$.

The results of the paper are probably valid and merits publication in Nature Communication journal. I have however one general concern.

This is a good piece of the experimental work in which the authors have discovered the dome-shaped structure of the magnetic phase in FeSe superconductor. They have shown that magnetism disappears for pressures exceeding 6GPa, and that such disappearance is associated with the sudden increase of the superconducting transition temperature T_c up to $\sim 38\text{K}$.

The results of the paper are probably valid and merits publication in Nature Communication journal. I have however one general concern.

The determination of the magnetic ordering temperature via observation of an anomaly on the resistivity curve is doubtful. Even in papers cited by the authors (e.g. Ref.18) it states that the quantitative agreement between T_N determined by μSR and T_m from resistivity experiments is very pure. In addition, the transition to the magnetic state obtained by means of muons is quite broad, while T_m anomaly on the resistivity curve is rather sharp -- why it is so?

I would like to see the bulk prove of direct relation between T_N and T_m . The authors could conduct, e.g., NMR (there is a group directly in IOP), neutron or μSR experiments on exactly the samples studied.

Relation with the Nature Communications criteria:

1. Originality and interest: if not novel, please give references

Referee: Results are novel.

2. Data & methodology: validity of approach, quality of data, quality of presentation

Referee: quality of presentation is very good. The validity of the main approach (the direct relation between T_n and T_m) need to be confirmed.

3. Conclusions: robustness, validity, reliability

Referee: Conclusions are probably valid.

4. Suggested improvements: experiments, data for possible revision

Referee: Additional bulk sensitive experiments are required.

5. References: appropriate credit to previous work?

Referee: Full credit.

6. Clarity and context: lucidity of abstract/summary, appropriateness of abstract, introduction and conclusions

Referee: The paper is clearly written.

Reviewer #3 (Remarks to the Author)

This manuscript reports on the comprehensive phase diagram of FeSe by utilizing magneto-transport measurements under high pressure. The obtained phase diagram highlights emergence of the magnetic phase supplanting the nematic phase and a sudden enhancement of superconducting transition temperature accompanied by the suppression of the magnetic phase at higher pressure. Reminiscent of high T_c cuprate superconductors, the observation of high T_c phase adjacent to the magnetic order in FeSe may provide a key ingredient for the exotic pairing origin in the iron based superconductors.

One possible improvement of this manuscript would be to provide the data on the pressure dependence of the slope of upper critical field at T_c , $(1/T_c)(dH_{c2}/dT)|_{T=T_c}$. Closely connected to the square of the quasiparticle effective mass (see, V. G. Kogan and R. Prozorov, *Rep. Prog. Phys.* 75, 114502 (2012)), this slope is expected to be enhanced at a QCP. Together with the authors' argument about the linear-in- T resistivity and unusual pinning, the data on the slope of H_{c2} will allow to make a good case for the presence of the QCP.

Overall, I find this manuscript intriguing. The results presented are sound, sufficiently novel, and important to the on-going debate concerning unusual superconductivity observed in iron pnictide class of materials to justify publication of this interesting study.

Reviewers' Comments:

Reviewer #1 (Remarks to the Author)

[The referee believes the revised version is now acceptable for publication with no further comments for the authors]

Reviewer #2 (Remarks to the Author)

I was not aware of experiments mentioned by the authors. With these references cited in the paper I would agree with the publication of manuscript at Nature Communication journal.

Reviewer #3 (Remarks to the Author)

The authors have provided a reasonable and satisfactory replay to my comments, and the manuscript has been improved. Therefore, the paper meets all the standards for publication in Nature Communications, and I recommend publication as it is.

First of all, we thank all the reviewers for taking their valuable time to review our manuscript. Below is our point-to-point response to the comments.

Response to Reviewer #1

The correlation between superconducting transition temperature T_c and electronic or spin orders such as spin or charge density wave or nematic ordering is the most important issue in both cuprate and Fe based superconductors. Therefore, the construction of phase diagrams for T_c in relation to a number of parameters such as charge carrier concentrations, magnetic transition temperatures, crystal lattice and symmetry, pressure or magnetic field is crucial to uncover what is closely linked to T_c . Among Fe-superconductors, FeSe superconductor's T_c in relation to magnetism has been unclear. By performing transport study at high pressure using three different pressure approaches, the authors have observed an evolution of T_c in relation to T_s (nematic transition) and magnetic transition, T_m . The authors constructed phase diagram which shows that dome-shaped magnetic order competes with the superconductivity of FeSe under high pressure. If the diagram reflects the T_c 's intrinsic correlation with T_s or T_m , it would be quite interesting, unique, and useful for further understanding of the high T_c superconductivity in FeSe. To achieve the intrinsic correlation, ideally, the crystal structures or crystal symmetry should remain unchanged at various high pressures. According to previous studies (EPL, 99 (2012) 26002; J Phys Chem B. 2010 Oct 7;114(39):12597-606. doi: 10.1021/jp1060446. Etc), high pressure can cause a few crystal structural or symmetry changes in FeSe.

The following are comments and suggestions:

1) The authors need to take into account the pressure induced possible structural changes and discuss how they can affect the phase diagram.

We appreciate the reviewer's acknowledgements of the possible importance of our work. To address his/her valid concern on the effect of structural changes at high pressure, we have performed additional measurements up to 15 GPa. In the papers introduced by the reviewer, which we now cite in the revised manuscript, it has been shown that the room-temperature crystal structure changes from tetragonal $P4/nmm$ to collapsed orthorhombic $Pbnm$. This high-pressure collapsed orthorhombic phase is different from the low-temperature orthorhombic $Cmma$ in the nematic phase at ambient pressure, and exhibits no superconductivity. Our new data, taken by using a small (2.5 mm) cubic anvil cell, show that the temperature dependence of resistivity changes from metallic and superconducting behaviour at < 12 GPa to semiconducting and non-superconducting one at high pressures above 12 GPa. This is consistent with the pressure-induced crystal structure change reported in the mentioned papers. We also note that in the early stage, the contamination of the secondary hexagonal phase in polycrystalline samples are reported and at high pressures the volume fraction of this secondary hexagonal phase increases. However, this hexagonal phase is also known to be

non-superconducting. Our high-quality single crystals do not have this secondary phase. Moreover, the very recent X-ray work under high pressures up to 4 GPa using single crystals (arXiv1603.04135) confirmed that the crystal structure is tetragonal, which transforms to the orthorhombic structure below T_m as in the nematic phase in the ambient pressure. In the pressure range up to 11 GPa we have observed systematic pressure dependence of metallic resistivity curves. From these results, we can safely conclude that the superconducting phase diagram obtained below 11 GPa is for FeSe having the same structure symmetries as those at ambient pressure.

2) The authors define the T_c using zero resistance temperature. However, the superconducting transition temperature (T_c) seems more appropriate than $T_c(\text{zero})$ as a transition T_c may appear far above zero resistance T_c . For example, in Fig.2 b, the T_c (zero resistance is low), however, a transition T_c may already appear between T_m and T_c (zero resistance) for pressures from 2.8-4.8 GPa, which becomes very clear for 5.8 GPa. If this is the case, the shape of T_c vs pressure could be quite different in the phase diagram (Fig.1).

3) To clarify if the transition T_c is real, magnetic measurement under high pressure seems necessary.

To address the issue of the determination of T_c , we have performed ac susceptibility measurements under pressure. The temperature below which the susceptibility $\chi(T)$ starts to decrease is in excellent agreement with T_c determined by the zero resistivity (see new phase diagram in Figs. 1 and 5). The volume fraction estimated from $4\pi\chi$ values at the lowest temperatures does not have strong pressure dependence, suggesting that the bulk nature of superconductivity continues in the entire pressure range up to 9 GPa.

4) The long tail of resistance from 2.8 to 5.8 GPa in Fig.2 may indicate the coexistence of secondary phase induced by pressure.

This log tail only appears in the coexistence region between magnetism and superconductivity. It has been often reported, in e.g. BaFe_2As_2 -based materials, that the superconducting transition becomes broad in the coexistence region. This may be because the Fermi surface is small in the antiferromagnetically ordered phase, leading to the reduced number of pairing electrons, which could enhance the fluctuation effect. This is supported by the large regime of vortex liquid state in the H - T phase diagrams in the coexistence region (Fig. 4). We also note that the inevitable inhomogeneity of pressure in the sample may cause additional broadening when the pressure dependence of T_c is steep, which can be seen in the 5.8 GPa data.

5) The selenium concentrations could be different for crystals made by CVD and flux growth method. This can give different crystal structural changes under high pressure.

As the transport properties are sensitive to the crystal quality and nonstoichiometry, we compared the physical properties of the crystals grown using different techniques. As stated in the METHODS section, crystals grown by vapour transport technique at Kyoto University and those by flux method at Oak Ridge National Laboratory have similar phase transition temperatures ($T_s = 87 - 90$ K and $T_c = 8.5 - 9$ K) and similar residual resistivity ratio $RRR \sim 40$ at ambient pressure. The comparison confirms that flux growth and vapor transport crystals used in this study are of similar quality. The compositional difference, if any, should not affect the response to pressure of all transitions of interest. This is confirmed by the excellent agreement between phase diagrams obtained in the University of Tokyo and Institute of Physics.

6) If possible, crystal structural determination under the same pressure range should be done for the same crystal samples. This is critical to find out if it is the structural change or T_m induced by pressure that is responsible for the changes of T_c in FeSe.

As mentioned above, the pressure-induced superconducting to semiconducting change has been observed, which is associated with the crystal structure change reported before. The detailed crystal structure determination in the high-pressure semiconducting phase deserves further studies but we believe that this is beyond the scope of the present paper. We also note that during the review process, several related preprints appeared in arXiv.org, and now the crystal structure in the magnetic phase has been shown to be orthorhombic, similar to the nematic phase (arXiv1603.04135).

Response to Reviewer #2

This is a good piece of the experimental work in which the authors have discovered the dome-shaped structure of the magnetic phase in FeSe superconductor. They have shown that magnetism disappears for pressures exceeding 6GPa, and that such disappearance is associated with the sudden increase of the superconducting transition temperature T_c up to 38K.

The results of the paper are probably valid and merits publication in Nature Communication journal. I have however one general concern.

The determination of the magnetic ordering temperature via observation of an anomaly on the resistivity curve is doubtful. Even in papers cited by the authors (e.g. Ref.18) it states that the quantitative agreement between T_N determined by μ SR and T_m from resistivity experiments is very pure. In

addition, the transition to the magnetic state obtained by means of muons is quite broad, while T_m anomaly on the resistivity curve is rather sharp – why it is so?

I would like to see the bulk prove of direct relation between T_N and T_m . The authors could conduct, e.g., NMR (there is a group directly in IOP), neutron or μ SR experiments on exactly the samples studied.

We thank Reviewer #2 for the positive statements for publication in Nature Communications. The μ SR measurements have been reported for polycrystalline samples (Ref. [22] in the revised manuscript), which likely exhibit large inhomogeneity in the chemical composition. In the same paper, the authors state that the neutron scattering cannot detect any magnetic signals, indicating small magnetic moment. We have also tried neutron scattering measurements under pressure in our single crystals, but we could not observe directly the magnetic order within the resolution, consistent with the small moment (please note that neutron intensity is proportional to the square of the magnetic moment). During the review process, two preprints appeared reporting evidence for antiferromagnetic order below T_m from Mössbauer (arXiv1603.04135) and NMR (aXiv:1603.04589), both of which are sensitive probe for small magnetic moment. The phase diagrams for the magnetic order up to 4 GPa reported in these preprints are quantitatively consistent with our phase diagram. These results, together with the sharp change in the resistivity indicate that the pressure-induced magnetic transition is of first order at least up to 4 GPa.

Response to Reviewer #3

This manuscript reports on the comprehensive phase diagram of FeSe by utilizing magneto-transport measurements under high pressure. The obtained phase diagram highlights emergence of the magnetic phase supplanting the nematic phase and a sudden enhancement of superconducting transition temperature accompanied by the suppression of the magnetic phase at higher pressure. Reminiscent of high T_c cuprate superconductors, the observation of high T_c phase adjacent to the magnetic order in FeSe may provide a key ingredient for the exotic pairing origin in the iron based superconductors.

One possible improvement of this manuscript would be to provide the data on the pressure dependence of the slope of upper critical field at T_c , $(1/T_c)(dH_{c2}/dT)|_{T=T_c}$. Closely connected to the square of the quasiparticle effective mass (see, V. G. Kogan and R. Prozorov, Rep. Prog. Phys. 75, 114502 (2012)), this slope is expected to be enhanced at a QCP. Together with the authors' argument about the liner-in-T resistivity and unusual pinning, the data on the slop of H_{c2} will allow to make a good case for the presence of the QCP.

Overall, I find this manuscript intriguing. The results presented are sound, sufficiently novel, and important to the on-going debate concerning unusual superconductivity observed in iron pnictide class of materials to justify publication of this interesting study.

We are grateful to see the Reviewer #3's recommendation for publication of our study. As suggested by the reviewer, we have added an analysis of the H_{c2} slope (see Fig. 6). The results do not show a dramatic change near the possible QCP, but similar non-singular dependence of H_{c2} has also been reported across the QCP in $\text{BaFe}_2\text{As}_{1-x}\text{P}_x$. As the anomalous vortex state is expected near the QCP where the antiferromagnetism may affect the vortex cores, it has been discussed that how H_{c2} should look like across the QCP is a nontrivial issue (Ref. [40] in the revised manuscript). This could stimulate further experimental and theoretical studies.